

# Development of Internet-of-Things-Based Controlled-Source Ultra-Audio Frequency Electromagnetic Receiver

Zucan Lin[1], Qisheng Zhang[1], Keyu Zhou[1], Xiyuan Zhang[1], Xinchang Wang[1], Hui Zhang[1], and Feng Liu[1]

[1]School of Geophysics and Information Technology, China University of Geosciences (Beijing), Beijing, China

**Correspondence:** Qisheng Zhang (zqs@cugb.edu.cn)

**Abstract.** Electromagnetic exploration, characterized by its low cost, wide applicability, and high operational efficiency, finds extensive applications in fields such as oil and gas exploration, mineral prospecting, and engineering geology. Traditional controlled-source electromagnetic detection methods are typically confined to operating frequencies below 250 kHz, resulting in insufficient detection accuracy for applications such as shallow and intermediate-depth exploration, thereby constraining

their performance in high-resolution imaging. To address these challenges, we propose a controlled-source ultra-audio frequency electromagnetic receive system based on the Internet of Things (IoT). We investigate cascaded digital filtering and sampling techniques to extend the receiver's sampling rate range, thereby elevating the operating frequency of controlled-source electromagnetic acquisition from the conventional maximum of 250 kHz to 1 MHz. The receiver achieves a sampling rate of up to 2.5 MHz, comprising three magnetic field measurement channels and two electric field measurement channels.

The instrument is compact, lightweight, and capable of real-time data storage locally, and real-time data transmission to an upper computer. Additionally, IoT technology is introduced, leading to the design of a cloud-based real-time remote control and data acquisition scheme. Experimental results demonstrate the stability of the instrument, meeting the requirements of field exploration.

## 1   Introduction

Electromagnetic exploration, renowned for its low cost, widespread applicability, and operational efficiency, is extensively utilized in fields such as oil and gas exploration, mineral prospecting, and engineering geology (TENG et al., 2022; WANG et al., 2022; Chun-lei et al., 2022; Zhou et al., 2021a; Wang et al., 2023). The electromagnetic exploration method (EM) can be categorized into time-domain electromagnetic method (TEM) and frequency-domain electromagnetic method (FEM). The FEM (Peng et al., 2024)primarily encompasses magnetotellurics (MT), which employs natural field sources, and controlled-

source magnetotellurics (CSMT), which utilizes artificial field sources. CSMT is commonly applied in geothermal resource exploration (Aykaç et al., 2015; Zhang et al., 2022), mineral resource exploration, hydrological surveys, engineering geology, and other fields (Tang and Wang, 2023; Liu et al., 2022; Guo et al., 2020; Farzamian et al., 2019). CSMT method can be classified into controlled-source audio-frequency magnetotellurics (CSAMT) and controlled-source radio-frequency magnetotellurics (CSRMT) based on different operating frequencies. CSAMT (Zhou et al., 2021b) typically operates within the

frequency range of 0.1 Hz to 10 kHz, with exploration depths ranging from tens of meters to two to three kilometers, making





it suitable for exploring geothermal and mineral resources in the subsurface. CSRMT (Xu et al., 2014) operates within the frequency range of 10 kHz to 250 kHz. Traditional controlled-source electromagnetic detection methods are usually limited to operating frequencies below 250 kHz, which poses challenges in achieving adequate detection accuracy for applications such as shallow and intermediate-depth exploration, thereby restricting its performance in high-resolution imaging.

Additionally, a single transmission source and a single-directional transmission strategy are typically employed in existing controlled-source electromagnetic exploration methods. This makes them susceptible to shielding effects from underground resistive bodies, limiting the comprehensive perception of electromagnetic field signals and thereby reducing the effectiveness of exploration. Moreover, due to the use of a single-directional transmission method, transmission efficiency is low, further impacting the accuracy of exploration. Therefore, the presence of these issues urgently necessitates an innovative controlled-
source electromagnetic exploration method to enhance operating frequencies, improve detection accuracy, and address the obstacles to efficient exploration present in traditional methods.

Simultaneously, the electromagnetic receivers widely used in exploration commonly suffer from low sampling rates and typically utilize data storage media such as SD cards, which cannot meet the demands for multi-channel, high sampling rate, and long-term continuous data acquisition. To address these issues, we propose a controlled-source ultra-audio frequency
electromagnetic receiver with a working frequency range of 1 Hz to 1 MHz, which can further improve accuracy in shallow to intermediate-depth exploration. The device is portable and capable of instantly storing data locally, while also supporting real-time data transmission to a central processing unit. Additionally, by integrating Internet of Things technology, we have developed a cloud-based solution that enables real-time remote control of the device and data collection functionalities.

## 2   Basic principles

The Magnetotellurics method (MT) was first proposed by the French scholar Cagniard in 1953 (Cagniard, 1953). CSMT evolved from Audio Magnetotellurics (AMT). In 1975, Goldstein and Strangway introduced the use of artificial electrical sources in AMT and discussed their application in mineral exploration (Goldstein and Strangway, 1975). The emergence of CSMT has addressed the instability of the field source and the difficulty in obtaining high signal-to-noise ratio data in AMT, greatly advancing the development of electromagnetic exploration. However, it has also introduced a series of source-related
issues such as susceptibility to terrain effects and non-planar wave effects.

A typical schematic diagram of a CSMT device is shown in Figure 1, and the scalar apparent resistivity calculation method is represented by Formula 1 (Zhang et al., 2021; RONG and LIU, 2022; Yu et al., 2023). Here, E and H represent the mutually perpendicular horizontal components of the electric and magnetic fields. In AMT, the field source is the electromagnetic field excited by distant lightning, which can be approximated as a plane wave in the exploration area. However, when artificial
sources are used for exploration, as the sources are closer to the survey area, not all electromagnetic waves in the exploration area can be simply regarded as plane waves. In 1982, Sandberg described the distribution of electromagnetic fields excited by artificial sources in a uniform half-space and pointed out that the electromagnetic waves at a position could be considered as plane waves, and the apparent resistivity calculated according to Formula 1 could approach the true resistivity, only when the



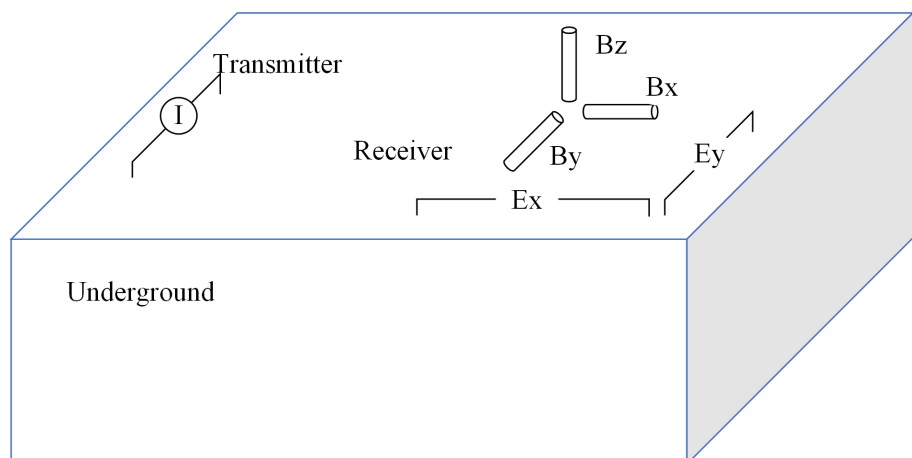

**Figure 1.** Schematic diagram of CSEM

receiver-transmitter distance was greater than 3 times the skin depth (receiver located sideways to the transmitter) or 5 times the skin depth (receiver located in the same direction as the transmitter). This phenomenon is known as the near-field effect (Sandberg and Hohmann, 1982). There are typically two ways to avoid the near-field effect: one is to deploy stations reasonably to exclude data from the near-field area; the other is to correct data using near-field correction algorithms to achieve full-area detection.

$$\rho_\alpha = \frac{1}{\mu\omega} \frac{|E|^2}{|H|^2} \tag{1}$$

## 3   Design of IoT based Controlled-source Ultra-audio Frequency Electromagnetic Receiver

### 3.1   Overview of the architecture

The overall hardware structure of the receiver is illustrated in Figure 2, comprising the interface board, analog board, connection board, and main control board. The interface board serves as the bridge between the internal and external components of the instrument, integrating sensor interfaces, power interfaces, communication interfaces, indicator light interfaces, and keypad interfaces. The sensor interfaces consist of magnetic field sensors and electrode terminals, with their connections linked to the analog channel input terminals on the analog board. The power interface is responsible for supplying external 12V power, which is distributed through the power distribution network to the entire system. The communication interface, indicator lights, and keypad facilitate human-machine interaction, with their internal wiring connected to the Multiprocessor System-on-Chip (MPSoC) on the main control board. Additionally, the interface board has functions such as overcurrent protection and battery monitoring. The digital interfaces of the power monitoring chip and power control chip on the interface board are also connected to the MPSoC on the main control board.





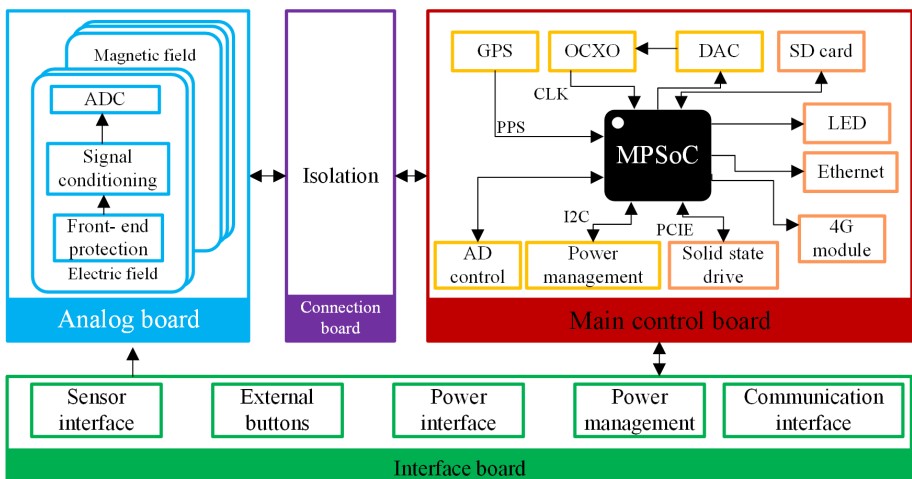

**Figure 2.** The overall architecture of the receiver

On the connection board, there is a magnetic coupling isolation module, which connects to the digital signal ports of both the analog board and the main control board. It is responsible for isolating the digital and analog parts, severing electrical connections to reduce interference from the digital part on the analog signals.

## 3.2 Design of the analog board


The analog board is designed with five analog channels, each responsible for conditioning and analog-to-digital conversion (ADC) of signals from two electric field components and three magnetic field components. As shown in Figure 3, the signal conditioning circuit mainly consists of input protection circuits, programmable gain amplifiers, single-ended to differential conversion circuits, and anti-aliasing filtering circuits. We employ the AD8253 amplifier (Yuan et al., 2016) as the programmable

gain amplifier, which boasts a high slew rate of up to 20 V/$\mu$s, meeting the bandwidth requirements for ultrasonic data acquisition. It provides programmable gains from 1 to 1000, enhancing the circuit's capability to capture weak signals. The analog-to-digital conversion circuitry includes the AD7760 (Zhang et al., 2015) analog-to-digital converter and its peripheral circuitry, with its digital interface connected to the MPSoC on the main control board via the connection board. The AD7760 is capable of outputting 24-bit precision sampling data at a maximum frequency of 2.5 MHz.

## 3.3 Design of main control board


We employ the XCZU3EG MPSoC as the primary control chip (Khandelwal and Shreejith, 2022), and the physical appearance of main control board is depicted in Figure 4. Internally, XCZU3EG consists of two internal components: ARM and FPGA with distinct responsibilities respectively. The FPGA portion primarily manages the analog board for ADC, receives and parses converted data, processes GPS data, calibrates Oven Controlled Crystal Oscillator (OCXO), and controls system power. On the

other hand, the ARM portion handles the overall workflow control of the entire acquisition station, human-machine interaction,





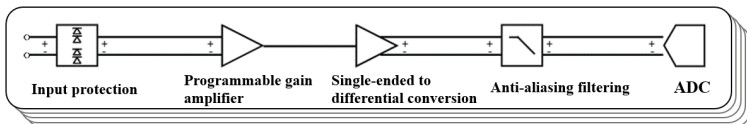

**Figure 3.** Block diagram of analog channels

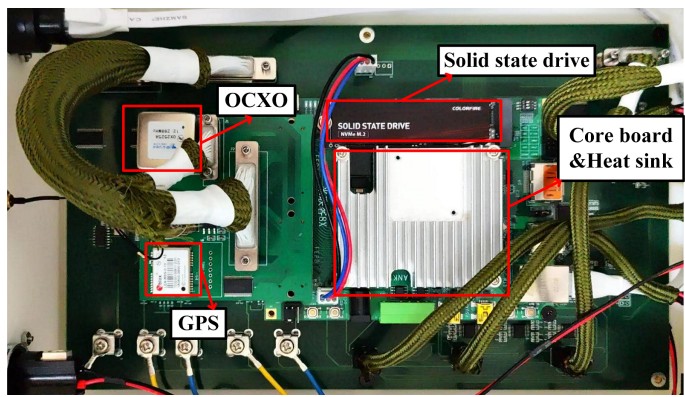

**Figure 4.** Image of main control board

and data storage. Communication between the ARM and FPGA sections occurs via the internal AXI bus, allowing the ARM part to configure acquisition parameters and control acquisition start/stop by accessing FPGA registers. After parsing by the FPGA, the analog-to-digital converted data is transmitted to the ARM section in the form of data streams, stored onto external solid-state drives, and transmitted in real-time to the host computer via Ethernet. The ARM part also manages functions related

to buttons, LEDs, and facilitates remote real-time monitoring through data transmission with a 4G module. Additionally, the main control board includes circuits for storing program code on a Micro-SD card, storing acquisition data on solid-state drives, communication via gigabit Ethernet, time synchronization via GPS, and providing stable clock signals through OCXO. The solid-state drives offer read/write speeds of up to 3.5 GB/s, meeting the requirements for real-time storage of multi-channel high-sampling-rate full waveforms in ultrasonic data acquisition (40 MB/s). The OCXO-related circuits consist of a DAC

circuit and a voltage-controlled OCXO, where the DAC generates a voltage signal to calibrate the voltage-controlled OCXO.

## 3.4 4G module

To ensure efficient and reliable data transmission and remote management in our system, we have selected the cost-effective USR-G771-GL 4G DTU module (Zhang and Wang, 2022), manufactured by USR IoT. This module supports multiple 4G LTE frequency bands worldwide, ensuring extensive network coverage and compatibility. Moreover, it offers high-speed data

transmission and extremely low communication latency, enabling real-time data processing and rapid response. The USR-G771-GL module is an ideal choice for the digital transformation and network expansion of our system, meeting the stringent requirements of modern industry for high-speed, stable, and remote communication.





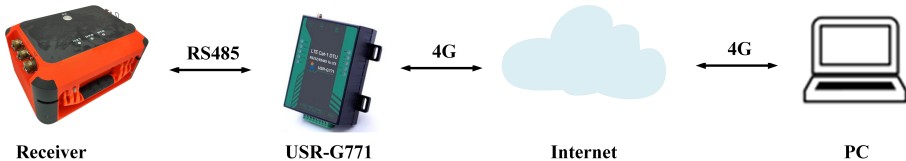

**Figure 5.** Connection topology

Equipped with standard serial port interfaces, the module seamlessly connects to the main control unit, facilitating seamless data exchange. This simple connectivity greatly promotes the intelligent upgrading of traditional serial port devices, provid-
ing them with wireless network access capabilities. Whether in industrial automation, intelligent transportation systems, or environmental monitoring, the module demonstrates outstanding performance.

By employing the USR-G771 module, users can achieve remote monitoring and management of the receiver, enabling centralized data collection and control regardless of the receiver's location. This capability significantly enhances operational efficiency, reduces maintenance costs, and increases system flexibility. The connection topology is illustrated in Figure 5.

**3.5 Programming design**

**3.5.1 Lower computer program framework**

The FPGA part in the MPSoC is a programmable logic device and serves as the main component of the digital circuitry in the receiver. The program structure of the FPGA is expected to include various modules, as illustrated in Figure 6.

Data acquisition module is responsible for configuring the ADC's acquisition parameters, controlling ADC start/stop, re-
trieving and parsing ADC data. It consists of five identical ADC control sub-modules, each handling data acquisition for one channel. The data from the five channels is stored in separate FIFO memories, processed, and then sent to the DMA control module for transmission to the ARM part. Additionally, this module controls the gain of the programmable amplifier.

DMA control module handles the transfer of collected data to the ARM part. It includes a FIFO memory. When the number of collected data exceeds a predefined threshold, the DMA module initiates data transfer. The data is transmitted directly to
the DDR memory on the ARM side via the AXI4-Full bus. After completing the set number of transfers, the DMA module generates an interrupt. Upon receiving this interrupt, the ARM processor retrieves the data from the DDR memory, switches the DMA storage target address, reconfigures the DMA module via the AXI4-Lite bus, and starts the next round of data transfer.

Oscillator calibration module is responsible for calibrating the Oven Controlled Crystal Oscillator (OCXO) to provide accurate clock signals for the entire FPGA section. It includes a DAC control sub-module and a frequency measurement sub-module.
The frequency measurement module measures the frequency of the OCXO signal after multiplication by the GPS second pulse signal. Based on the difference between the measured frequency and the standard frequency, the DAC control sub-module adjusts the voltage signal to regulate the OCXO frequency until the difference falls within a certain range, completing the calibration.


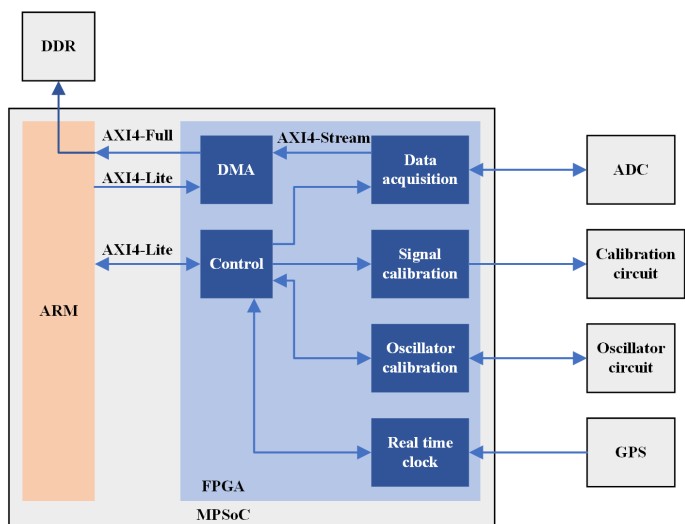

**Figure 6.** FPGA program structure diagram

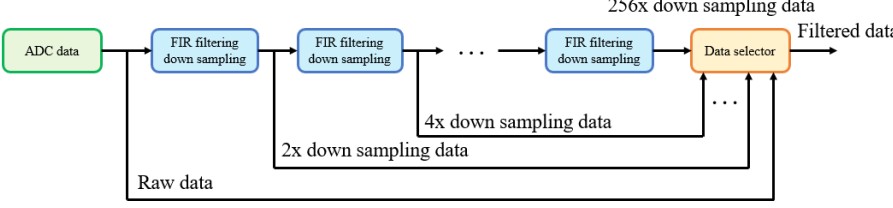

**Figure 7.** Schematic diagram of cascaded digital filtering sampling

Real-time clock module receives serial data and second pulse signals from GPS, parses the serial data to obtain the current

time, and synchronizes the system time based on the rising edge of the second pulse. After synchronization, the real-time clock module continuously generates local time using the calibrated local clock signal for the entire system.

Signal calibration module generates square wave signals at specific frequencies to provide reference signals for the calibration circuitry of the analog board. It divides the calibrated local clock signal and outputs it as a reference signal, with the division ratio configured by the instruction control module.

Instruction control module controls and configures the other modules. The program in the ARM part can read and write a series of registers in the instruction control module via the AXI4-Lite bus to control the operation of the other FPGA modules.

### 3.5.2 Cascade Digital Filter Sampling Technology

To meet the wide-bandwidth requirements of the receiver, the cascade digital filtering and sampling function is implemented in the FPGA part of the MPSoC. Its structure, as depicted in Figure 7, consists of cascaded stages, each composed of a 96-tap FIR

filter, providing 120 dB attenuation at the Nyquist frequency. Cascade digital filtering and sampling prevent spectrum aliasing



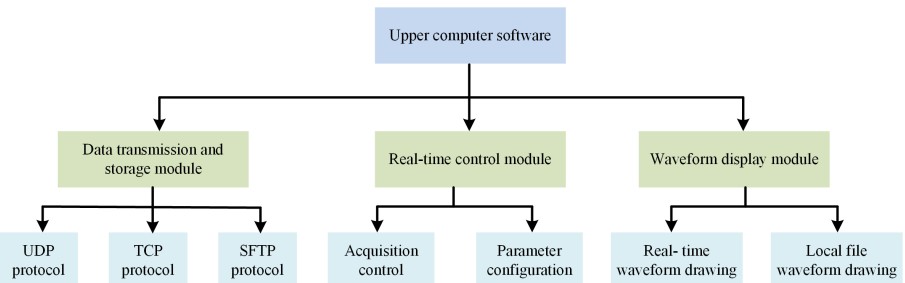

**Figure 8.** Block diagram of PC-end software

**Table 1.** Comparison of performance indicators of electromagnetic receivers

| Instrument model | Stratagem EH5 | V8 | GDP-32II | ADU-07 | Developed Receiver |
| --- | --- | --- | --- | --- | --- |
| Manufacturer | Geometrics | Phoenix | Zonge | Metronix | China University of Geosciences |
| Sampling rate | 75 Hz-192 kHz | 96 kHz | 32 kHz | 512 kHz | 305 Hz-2.5 MHz |
| Frequency range | 10 Hz -96 kHz | 0.00005-10 kHz | 1/64 Hz-8 kHz | DC-250 kHz | 1 Hz-1 M Hz |
| Number of channels | 5 | 6 | 1 to 16 | 1 to 10 | 5 |
| ADC bits | 32 | 24 | 16 | 24 | 24 |
| Dynamic range | 127 dB | / | 190 dB | 130 dB | 143 dB |
| Power consumption | <8W | 15 W | / | <5 W | 10 W |
| Synchronization method | GPS | GPS | Quartz crystal /GPS | GPS | GPS+OCXO |
| Storage medium | Local storage | Local storage | Local storage | Local storage | Local storage + Remote transmission |
| Weight | 5.8 kg | 7 kg | 13.7 kg | 7.1 kg | 5.7 kg |

effects during down-sampling and suppress out-of-band quantization noise generated by the ADC process. Through cascade digital filtering and sampling technology, the sampling rate range of the analog-to-digital converter is expanded from 78 kHz to 2.5 MHz to 305 Hz to 2.5 MHz, ensuring the collection accuracy while allowing the instrument to be more flexibly applied in electromagnetic exploration across various frequency bands.

### 155   3.5.3   Upper computer program design

The upper computer mainly performs functions such as data reception and storage, waveform display, and instrument control. The functional architecture of the upper computer software is shown in Figure 8.

When using use the instrument, GPS synchronization and network connection is the first step. And then configure the acquisition frequency and time table to collect data. The overall workflow of the receiver is shown in Figure 9.





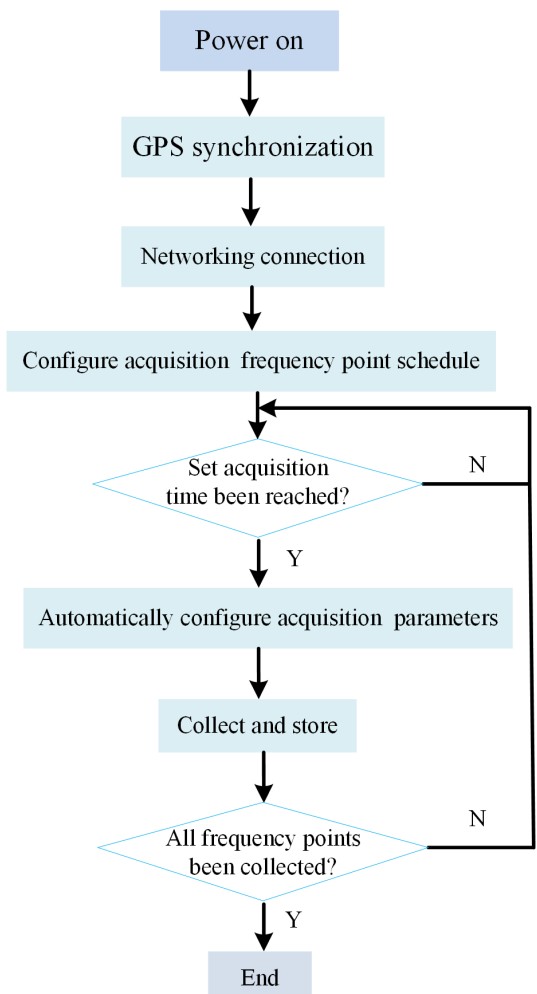

**Figure 9.** Workflow of PC-end software

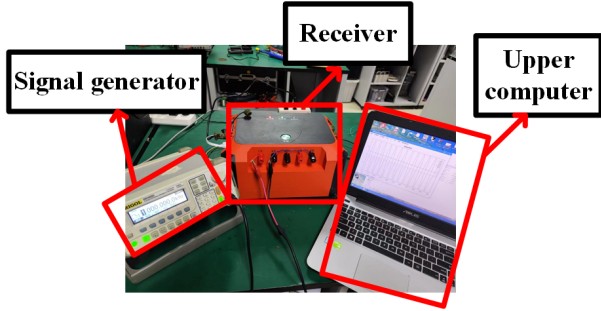

**Figure 10.** Photo of the testing site in the lab



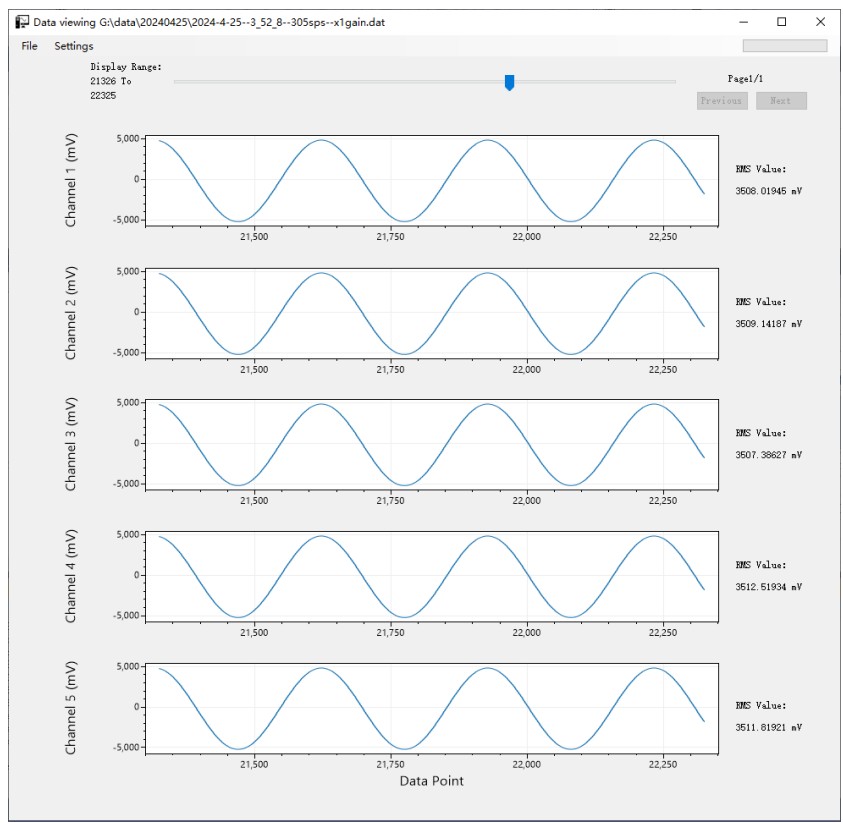

**Figure 11.** Measured value of the maximum undistorted sine wave

**Table 2.** 1Hz to 1MHz Signal attenuation result table

| Frequency (Hz) | Voltage amplitude (V) | Attenuation factor (dB) |
| --- | --- | --- |
| 1 | 1.00076 | 0.0066 |
| 10 | 1.00044 | 0.0038 |
| 100 | 1.00078 | 0.0068 |
| 1000 | 0.99885 | -0.010 |
| 10000 | 0.99467 | -0.046 |
| 100000 | 0.99617 | -0.033 |
| 600000 | 0.96660 | -0.30 |
| 1000000 | 0.95021 | -0.44 |



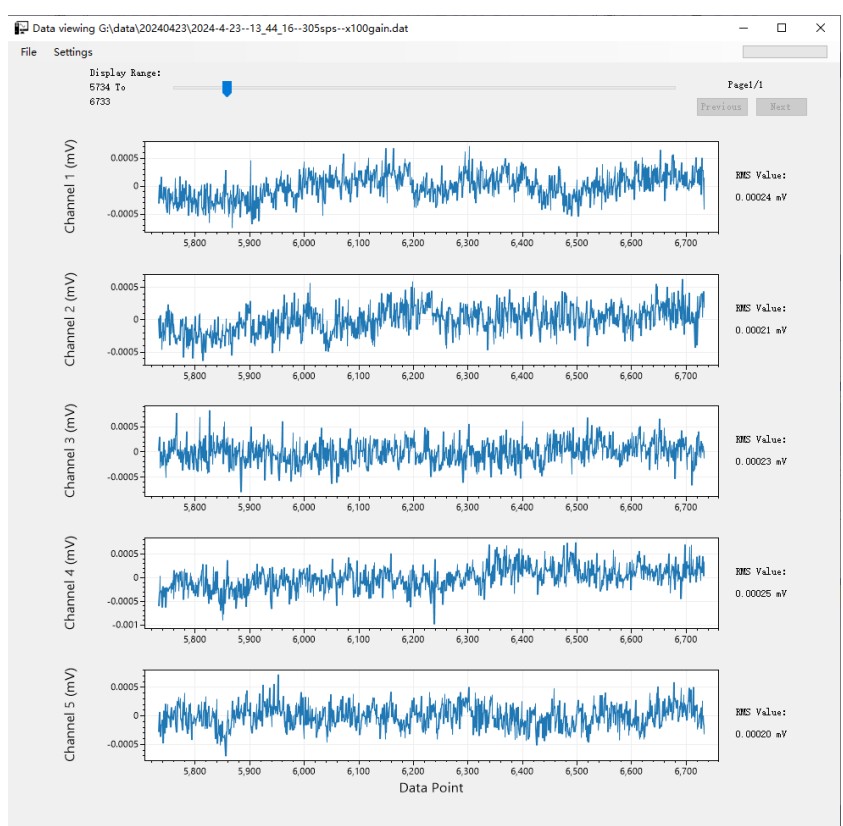

**Figure 12.** Measured value of short-circuit noise

## 3.6 Comparison of key instrument indicators


Some of the internationally advanced electromagnetic receivers include the GEOMETRICS EH5 from the United States (Geometrics, 2024), Metronix ADU07 from Germany (Metronix, 2024), Zonge GDP-32II from the United States (International, 2012), and Phoenix V8 from Canada (Geophysics, 2023). Partial parameters of these four instruments are presented in Table 1. From the table, it can be observed that current electromagnetic receivers already cover a wide frequency range. Particularly, the

Metronix ADU07 from Germany is a full-band magnetotelluric instrument, covering frequencies from direct current to radio frequency, enabling various detection methods such as magnetotellurics and controlled-source electromagnetic methods. Additionally, current electromagnetic receivers are moving towards multi-channel synchronous acquisition. All four instruments listed in Table 1 have at least two electric field channels and three magnetic field channels. The German Metronix ADU07 utilizes a combination of high-frequency and low-frequency dual channels, with a total of 10 channels, providing the most

detailed electromagnetic data for subsequent data processing. However, the sampling rates of these instruments do not exceed 512 kHz. In terms of storage media, they all use SD cards or other removable flash memory for data storage, and the speed of the storage media may become a bottleneck for high-speed continuous acquisition.





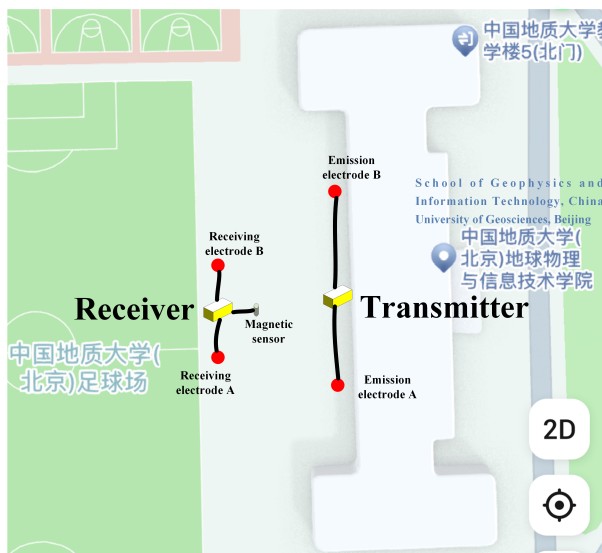

**Figure 13.** Outdoor test diagram (©Baidu map)

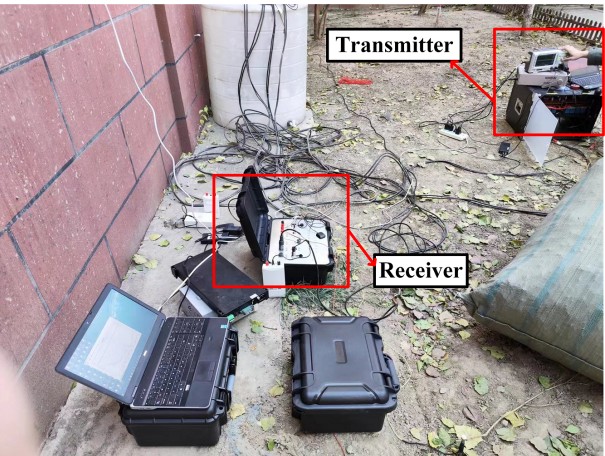

**Figure 14.** Image of outdoor testing

## 4 Instrument testing and result analysis

### 4.1 Frequency range testing

After the development was completed, we conducted system integration testing and performance evaluation (as shown in Figure 10), including tests on frequency range, background noise, dynamic range, and others.

After powering on the prototype, we connected the receiver and signal generator. A sinusoidal wave with a peak-to-peak amplitude of 1V was applied to the input of the receiver. The test results are summarized in Table 2.




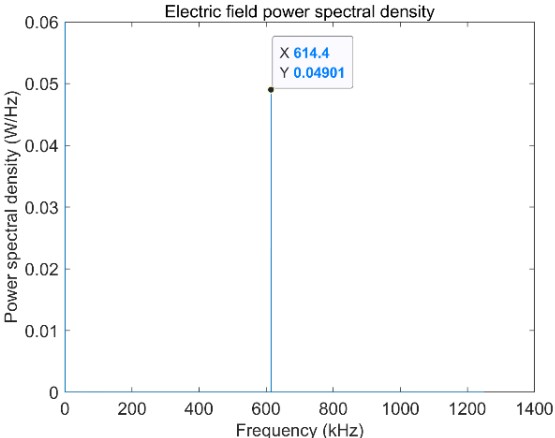

**Figure 15.** Electric field power spectral density

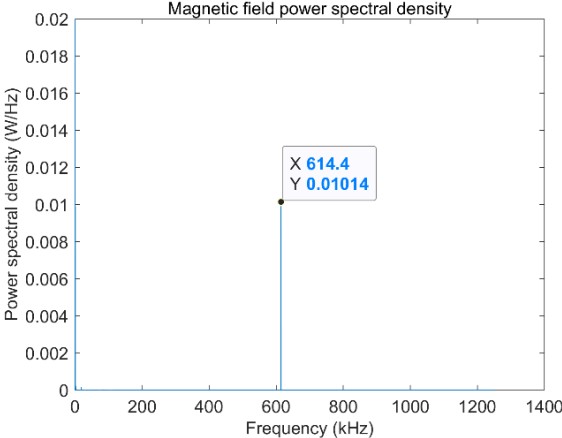

**Figure 16.** Magnetic field power spectral density

By varying the input signal frequency from 1 Hz to 1 MHz, it was observed that the attenuation factor remained below 3 dB.
This indicates that within the frequency range of 1 Hz to 1 MHz, the receiver operates within the -3 dB bandwidth.

### 4.2 Background noise and dynamic range testing

We remained the same connection setup, with the receiver inputting a sine wave with a peak-to-peak amplitude of 1 V. The input signal amplitude was gradually increased until the maximum undistorted sine wave was identified, resulting in a measured value of 3.5 Vrms, as depicted in the graph. When the receiver input was shorted, the effective value of the short-circuit noise
was measured, yielding a result of 0.25 $\mu$V, as shown in Figure 11 and Figure 12. Based on calculations, the dynamic range was determined to be around 143 dB.



## 4.3 Transceiver frequency test

We conducted a joint debugging test of transmitter and receiver outdoors. A 100 m cable was laid in the campus as shown in Figure 13, with a distance of 10m between the transmitting electrodes. Stainless steel electrodes and induction magnetic sensors
were used for reception, with a receiving cable of 50 m and a receiving distance of 10m. The outdoor test photos are shown in Figure 14. We use a transmitter to emit a 614.4 kHz square wave, and the receiver collects the spectrum of underground electric and magnetic field data as shown in Figure 15 and Figure 16. It can be seen that both the electric and magnetic field channels can clearly receive the 614.4 kHz signal, verifying the accuracy of the signal.

## 5 Conclusions

In this study, we have developed a novel IoT-based controlled-source ultra-audio frequency electromagnetic receiver and provided a comprehensive description of its software and hardware architecture. The system is compatible with both electrical and magnetic measurement techniques, with the receiver capable of sampling up to 2.5 MHz, effectively extending the operating frequency of controllable-source electromagnetic acquisition from the conventional 250 kHz to 1 MHz. This equipment is compact and portable, allowing for easy transportation, and it can instantly store collected data locally or transmit it in
real-time to higher-level devices such as computers. Furthermore, the system incorporates IoT technology, supporting remote real-time monitoring and control functions, significantly simplifying the complexity of field operations. Through experimental testing, we have validated the integrity of the system data, the efficiency of communication with the upper computer, and other performance indicators, all of which have met the expected design requirements.

*Data availability.* Our research is supported by national projects; thus, the data are not publicly accessible due to a confidentiality agreement.

*Author contributions.* This research was designed, tested, and implemented by the authors of the paper. The full text was designed and implemented by ZL. FL and KZ worked on the hardware design. XZ, ZL, and KZ worked on the software design. The other three authors (QZ, HZ, and XW) carried out revision and correction during the completion of the article, and they also performed the tests.

*Competing interests.* The authors declare that they have no conflict of interest.

*Acknowledgements.* This study is supported by the National Key R&D Program of China (Grant No.2022YFF0706202 and No.2021YFC2801404),
the National Natural Science Foundation of China (Grant NO.42074155) and the Key Research Program of the Chinese Academy of Sciences (Grant NO.KGFZD-145-22-06-02).



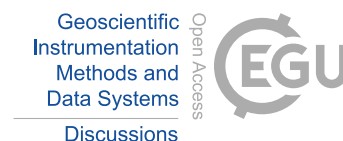

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
