# Peer review of "Development of Internet-of-Things-Based Controlled-Source Ultra-Audio Frequency Electromagnetic Receiver"

_Geoscientific Instrumentation, Methods and Data Systems, 2024_

## Author Comment (AC1)

26-08-2024

Dear reviewer,

We truly appreciate the time and energy you dedicated in carefully reviewing our manuscript.

Your comments were highly helpful. We really appreciate your attention and comments on our manuscript. Our replies are listed as follows:

*Comments 1.* Figure1: It appears that the Bx, By, Bz part of the figure is based on a left-handed coordinate system. Please check that this figure visually presents as intended.

Response 1: We appreciate you raising this point. We have modified Figure 1, changing Bx, By, and Bz to be based on the right-hand coordinate system. The modified image is shown below:

[Figure]

Fig1. Schematic diagram of CSEM

*Comments 2.* Line 80: The analog board is connected to both electric and magnetic field sensors. Please elaborate on the sensors and the necessary electrical protection and signal conditioning of the instrument.

Response 2: Thank you so much for your comment. The electric field sensor uses an active transmission line with a front-end amplifier, connected to a metal rod buried in the soil. The magnetic field sensor uses an inductive magnetic sensor. The input circuit for the signal is protected by Schottky diodes and current-limiting resistors. We have revised the corresponding section of the article. We have also pasted the modified paragraph below:

"As shown in Figure 3, the electric field sensor uses an active transmission line with a front-end amplifier, connected to a metal rod buried in the soil. The magnetic field sensor uses an inductive magnetic sensor. The input circuit for the signal is protected by Schottky diodes and current-limiting resistors as shown in Figure 4.

[Figure]

Fig3. Internal circuit of electrodes in electric field sensors

[Figure]

Fig4. Analog channel circuit"

*Comments 3.* Line142:It is not clear what the function of the signal calibration module is. Please add details on (the purpose of) the output reference signals.

Response 3: We appreciate you raising this point. The purpose of the signal calibration module is to calibrate the inductive magnetic field sensor. The inductive magnetic field sensor contains an internal coil that can generate a magnetic field. During calibration, our instrument outputs a

sinusoidal signal with a specific amplitude to this coil, generating a magnetic field of a certain magnitude. Our instrument then receives the response of the magnetic sensor to this magnetic field signal, allowing us to calibrate the sensitivity coefficient of the magnetic sensor at different frequencies. This sensitivity coefficient is used to convert the electrical signals output by the sensor back into magnetic field signals during actual measurements. We have revised the corresponding section of the article.

*Comments 4.* Line 153: The sentence on the frequency range of the instrument is unclear. Please rephrase and avoid the three "to" right after one another.

Response 4: Thank you for your careful review of the manuscript. We have modified the sentence.

*Comments 5.* Line 158: "using use"?

Response 5: Thank you for your careful review of the manuscript. We have modified the sentence.

*Comments 6.* Line 163, table 1. I'm not familiar with the instruments presented in the table, but I find it a bit odd to see that the 16-bit Zonge-GDP-32II has a 190 dB dynamic range. This indicates that some further tricks are needed. Please clarify and discuss in paper.

Response 6: Thank you so much for your comment. The 190 dB dynamic range of the Zonge-GDP-32II likely does not refer to the instantaneous dynamic range but rather to the ratio of the maximum input voltage to the minimum detectable signal when considering the channel gain. According to its manual, the maximum input voltage is ±32 volts, and the minimum detectable signal is 0.03 µV, with gain adjustable from 1/8 to 65536. In fact, the dynamic range specification of our instrument is determined in the same manner. We have modified the corresponding section of the article.

*Comments 7.* Line 185, Figure 11/12. It would be appropriate to augment both figures with a frequency analysis of the recorded data to show the spectral purity of the sine waves (Figure 11) as well as the shape of the noise floor of the instrument (Figure 12).

Response 7: We are grateful for your suggestion. We have added the signal's spectrum analysis chart. The added image is shown below:

[Figure]

Fig11 Measured value of the maximum undistorted sine wave

.

[Figure]

Fig12. Measured value of short-circuit noise

*Comments 8.* Line 185. Please add more detail on the data and calculations from which the 143 dB dynamic range is determined. The theoretical maximum dynamic range of a 24-bit converter is 144 dB and I am a bit surprised to learn that this instrument is so close to the maximum.

Response 8: We appreciate your comments. When testing the dynamic range, we consider the ratio of the maximum input voltage to the minimum detectable signal, taking into account the channel gain. The gain is 60 dB. We have modified the corresponding section of the article.

*Comments 9.* Line 192, Figure 15/16.These figures can be merged into one figure, and they should be plotted with a logarithmic y-axis. This choice of axis formatting will make the noise floor visible, and the performance of the instrument can be much better assessed. It would also be appropriate to show actual data and discuss the quality of these.

Response 9: Your suggestion is highly valued. We have modified the figure, which is shown below:

[Figure]

Fig18. Electric field and magnetic field power spectral density

---

## Author Comment (AC2)

26-08-2024

Dear reviewer,

We truly appreciate the time and energy you dedicated in carefully reviewing our manuscript.

Your comments were highly helpful. We really appreciate your attention and comments on our manuscript. Our replies are listed as follows:

*Comments 1.* Line64: Explain what is μ and ω.

Response 1: Thank you for your careful review of the manuscript. μ represents the magnetic permeability of the medium, and ω represents the angular frequency of the electromagnetic wave. We have added in the paper.

*Comments 2.* Line 89: Is a 24-bit precision really necessary given the measurement and the errors induced by the acquisition system?

Response 2: Thank you so much for your comment. 24-bit resolution is necessary because, in practical use, the amplitude of the electromagnetic field signal can sometimes be only a few tens of microvolts, making 16-bit accuracy insufficient.

*Comments 3.* Line94: Do not confuse GNSS receiver with GPS; there are other constellations besides GPS. This remark applies to the entire article.

Response 3: Thank you for your careful review of the manuscript. We have replaced GPS with GNSS in the paper.

*Comments 4.* Line 118: Is there an interface for remote operation of the system? If so, could we have an example?

Response 4: Thank you for your suggestions. We have added the interface of the remote operating system to the paper. The modified image is shown below:

[Figure]

Fig1. Interface of the remote operating system

*Comments 5.* Line 201: Possibility of integrating embedded computations to limit bandwidth and energy consumption?

Response 5: Thank you for your suggestions. It is possible to transmit only the processed results, but we usually also pay attention to the raw data collected.

*Comments 6.* Line 203: For monitoring in constrained locations, is it feasible for the system to operate on battery power?

Response 6: Thank you for your comment. Yes, the entire system can be powered by a 12V lithium battery.